# Modelling of the Fine-Grained Materials Briquetting Process in a Roller Press with the Discrete Element Method

**DOI:** 10.3390/ma15144901

**Published:** 2022-07-14

**Authors:** Michał Bembenek, Magdalena Buczak, Kostiantyn Baiul

**Affiliations:** 1Faculty of Mechanical Engineering and Robotics, AGH University of Science and Technology, 30-059 Kraków, Poland; magdalenka.buczak@gmail.com; 2Technological Equipment & Control System Department, Z.I. Nekrasov Iron and Steel Institute, National Academy of Sciences of Ukraine, 49050 Dnipro, Ukraine; baiulkonstantin@gmail.com

**Keywords:** roller press, discrete element method, briquetting process, modeling, forming elements

## Abstract

By using the Altair^®^ EDEM™ software, which implements the discrete element method, modelling and further study of the processes occurring in the roller press’s deformation area were carried out. It was shown that the discrete element method makes it possible to accurately describe the phenomena occurring in the area of roller press deformation compared with the finite element method. Models of material compaction in a roller press are developed using calcium hydroxide (slaked lime) and copper ore concentrate. The developed model makes it possible to determine the process’s energy parameters and the material’s compaction characteristics, taking into account the characteristics of its constituent particles. It was shown that discrete element modelling could be used effectively to create roller presses that provide rational characteristics of the briquetting process, taking into account the properties of the material being briquetted and the operating modes of the equipment. The results of the studies provided the basis for the applicability of the development of the discrete element method for describing the phenomena occurring in roller presses and accelerating the design of press equipment and briquetting technological processes.

## 1. Introduction

Roller presses are widely used in modern industrial enterprises for fine-grained raw material consolidation [1,2,3,4,5]. A significant amount of research was conducted to study the relationships between the design parameters of the roller presses, their operation parameters, material compaction characteristics in the deformation region, etc. [6,7,8,9,10,11,12,13,14]. At the same time, the issues of studying the processes of contact interaction of the compacted material with the pressing surfaces of roller presses are relevant. Conditions of contact interaction determine a significant number of characteristics of the briquetting process: energy consumption parameters, the fine-fraction material compaction size and characteristics, the wear conditions of the working surfaces of the rollers, etc. The processes in the roller press compaction unit largely determine the machines’ design, their operational and technological performance, and the quality of the briquettes. Therefore, improving known methods and creating new ones for predicting and analysing the parameters of the briquetting process in roller presses is an important and topical actual task.

Currently, there are three basic theoretical methods of analysis of the fine-grained material compaction process in the deformation region of the roller press: the Johanson method, the slab method, and the finite element method (FEM), as well as their combinations. The Johanson method [15,16] was the first method developed to determine the compaction parameters of fine-grained and powder materials in roller pressing. Johanson’s model describes the stress function concerning geometric parameters and the law of plastic yield of a material that undergoes constant shear deformation between rollers. His theory is based on the fact that the process can be divided into three separate regions: slip, nip, and release. In the Johanson model, the geometric parameters of the forming elements are taken into account indirectly by replacing the profiled roller with a smooth one of the given diameter. This approach does not allow for providing a stress distribution at the deformation region. That introduces inaccuracy in determining the briquetting energy parameters and does not allow for investigating the processes on the contact surface of the rollers. In addition to the configuration of the pressing surfaces, Johanson’s model has several assumptions about the physics of the compaction process and considers the flat problem of the stress–strain state. In particular, this model does not consider the influence of strain rate and change in friction parameters during the compaction process. Despite this, practical studies using the Johanson model made it possible to determine the characteristic angular parameters of the compacting process in roller presses, especially the pressing angle. These characteristics are used in other mathematical models of the compaction (briquetting) process in the roller presses. Despite its shortcomings, and above all, the model describing only a simplified strain state, much modern work is developed based on Johanson’s method [17,18,19,20,21,22,23,24,25,26,27,28].

Initially, the slab method was developed to study the sheet metal forming process [29]. It was developed by Katashinsky, who applied it to the study of the compaction parameters of metal powders in smooth rollers. The slab method is an analytical model of compaction of fine-grained materials. It takes into account the physical and mechanical properties of the material and the equation of equilibrium of the selected volume of the material under the action of pressing forces in the deformation region. The advantage of this model is that it allows for determining the two principal stresses, in contrast to the Johansson model, which allows one to determine only the average stress in the material. This model can also consider such characteristics of fine-fraction materials as adhesion and friction between particles. In addition, it allows for predicting the relationship between the size of the seal, the compression pressure, and the stress in the compacted material; this opens the possibility of developing methods for predicting the properties of briquettes, for example, density and strength. The advantage of the slab method is that it allows, with a slight deviation, for taking into account the real geometry of the pressing surface of the rollers.

The disadvantages of the Katashinsky slab method are that it does not consider the change in the coefficients of friction in the material with increasing compaction pressure. In addition, they solve the flat problem of stress–strain analysis. However, the slab method continues to evolve, as shown among other things by Hryniewicz in the 1990s [30]. The reason is its flexibility and simplicity of software implementation.

The finite element method (FEM) is quite well developed and widespread in various fields of mechanics [31,32]. This method has some ability to model in detail the compaction of fine-grained material in the rollers and, as a result, improves the optimisation of the technological process and equipment. It looks pretty attractive because it allows one to study the behaviour of fine-fraction materials and tribological conditions. However, there is a problem that this method is currently not sufficiently programmatically and theoretically developed for the study of multicomponent mixtures. That applies to fine-grained mixtures in which various components, fractions, and binders are present, including those containing a liquid phase. The results of calculations using the finite element method largely depend on the specific experimental data to determine, which requires quite expensive specialised equipment. There are studies on the application of this method to studying the parameters of briquetting, which investigated the effect of the configuration of the forming elements on the energy parameters of the process and the stress distribution in the compacted material [33,34,35,36]; however, in the authors’ opinion, too much simplification was used in the presented works, e.g., rollers roll material spread out on a flat surface.

Each of the three methods mentioned above does not accurately describe the phenomena occurring in the densification system of the roller press. According to the authors, further development of these modelling methods for material consolidation will not significantly impact the description of the actual conditions during material briquetting. Therefore, modelling the phenomena in the roller press compaction system requires a new approach.

One of the modern and advanced methods of modelling and deformation processes research of fine-grained bulk materials is the discrete element method (DEM) [37,38,39,40,41,42]. The DEM method is significantly different from the FEM method. In fine particle materials briquetting and compacting process in a roller press, FEM has been considered an efficient modelling tool. FEM modelling was adopted to simulate the roller briquetting compaction process with plane strain two-dimensional cases [19,23,32,33,34,35,36] and with three-dimensional models during the roller compaction processes [43,44,45,46]. However, FEM has been limited to macro-modelling, where it is used for simulating the interaction between the rolls and briquetted material. The material, in this case, is treated as a continuum. DEM gives a look into the micro-mechanical interactions of the particles within the briquetted material. This modelling method can provide data on the motion of the particles within the briquetted (compacted) material, particle collision forces, energy loss spectra, and power consumption, which are the data important for improving roller presses efficiency and gaining more understanding of the briquetting process kinetics.

That is why this method was chosen to perform research in the article. In order to conduct research on the fine-fraction materials compaction processes, the Altair^®^ EDEM™ (Discrete Element Method (DEM) Software) was selected, which is widely used to model physical processes using discrete elements [47,48,49,50].

The research significance of this work lies in the fact that, for the first time, the DEM method was used to study the processes in the roller press deformation region. Previously, this method was used only to study the movement of material in the screw feeder of the roller press. The aim of this work was to study the influence of the properties of the compacted material on the contact interaction conditions with the pressing surfaces of the rollers and the energy and power parameters of the briquetting process. The research results will make it possible to develop an approach for determining the rational characteristics of the compaction process of fine-fraction materials in roller presses, taking into account the fact that these materials consist of a set of discrete particles.

## 2. Materials and Methods

The research aimed to model the briquetting process in the LPW 450 (Figure 1a) roller press and compare the simulation test results with the actual results. A system of asymmetric pressing in rollers with saddle-shaped briquette production was considered (Figure 1b). The geometry of the compaction unit is shown in Figure 1c, while the view of the briquettes produced in it is shown in Figure 1d.

The research involved two materials:Calcium hydroxide—mixture of calcium hydroxide 85.1% (slaked lime) and water 14.9% with a moisture content of 15.0%;Copper ore concentrate mixture—copper ore concentrate with 5% sulphite lye (dry mass) and a moisture content of 4.2%.

The research covers the simulation of material briquetting in the roller press for the peripheral speed of rollers: 0.1, 0.15, 0.2, and 0.3 m/s.

### 2.1. Briquetting Material Model

Creating a material model for copper ore concentrate and calcium hydroxide was based on determining the main physical characteristics of their particles. In order to simulate the process of compaction of the studied materials in rollers, the computer program Altair^®^ EDEM™ Classroom (DEM Solutions Ltd., Edinburg, UK) was used. Because of the limitation of the classroom version of the EDEM in terms of the number of simulated material particles (10,000 elements), the most suitable sphere diameter of 2 mm was found, which at the same time provided a sufficient amount of material for consolidation and did not cause overly large calculation errors. The Hertz–Mindlin, elastic hysteresis, and linear cohesion models, were used. The Hertz–Mindlin model was used to simulate the contact between the surfaces of the working rollers and the particles being briquetted. The contact proposed by the Hertz–Mindlin model is a spring contact [51]. The model of elastic hysteresis contact allows for taking into account plastic strains in the equations of material mechanics—the main model responsible for material consolidation; constant contact occurs when a given stress value is exceeded. It is used to model the contact between molecules. The linear cohesion model is an additional model that only modifies the normal force value by adding the normal cohesive force (also used for contact between molecules).

The initial density of the bulk material is a vital parameter of the material model, as it determines the density of the briquette formed during the compaction process. When using the discrete element method, the initial parameter for modelling bulk material is the proper density of its particles. Due to the differences between the geometric parameters of real particles of the material and the elements assumed in the model, the particle density value is determined using the bulk density of the material. For calcium hydroxide with a moisture content of 15%, the bulk density is 0.61 g/cm^3^. A mixture based on copper ore concentrate with sulfite liquor is 1.55 g/cm^3^.

The determination of the particle density of the test material was carried out by a series of simulations of filling the cylinder with these particles with the expectation of its complete filling, taking into account the bulk density of the material (Figure 2). For modelling, a cylinder was used with a volume of 10.0 cm^3^ and the following dimensions of the internal cavity: diameter *d* = 17.84 mm and height *h* = 40.0 mm. With a cylinder volume of 10 cm^3^, taking into account the bulk density, the mass of the material was 15.6 g for a mixture based on copper ore concentrate and 6.1 g for calcium hydroxide, respectively. As a result of repeated stimulation of filling the cylinder, model values of particle density for the studied materials were established—for calcium hydroxide: 2.1 g/cm^3^, for a mixture based on copper ore concentrate: 4.5 g/cm^3^.

In order to determine the value of the Young’s modulus of the studied materials in a compacted state, the bulk material compaction process simulation in a closed matrix was used. As a reference, an author’s laboratory study of the variability of the apparent Young’s modulus of bulk materials in the close matrix of a 20 mm diameter was used presented in the article [52]. These studies assumed the variability of Young’s modulus depending on the material compaction degree. When using the discrete element method, Young’s modulus of each other particle is constant, but the material behaviour changes with the increasing contact between the particles. These studies were a model of the material behaviour during its compaction in a closed matrix with a diameter of 20 mm.

When modelling material compaction in a mould, the punch position was controlled. In order to determine the material compaction degree, the values of the punch position when it touches the compacted material at the initial pressing moment and, further, its position when the maximum pressing force value is reached were used.

The required stresses in the compacted material were achieved by applying the appropriate pressing force to the punch (Figure 3). Table 1 shows the data for the studied materials, including the established values of Young’s modulus.

The cohesion energy density coefficient value was selected for the material when determining Young’s modulus in such a way as to ensure the correct contact of the particles while maintaining the possibility of their free movement relative to each other in the absence of forces acting on the particles.

### 2.2. Model of Material Briquetting in a Roller Press

In order to simulate the briquetting process, an asymmetric roller pressing surface calibration was adopted (Figure 1b). The surfaces of the forming elements allow obtaining the saddle-shaped briquette (Figure 1d). The diameter of the rollers was 450 mm. There were 45 forming elements (cavities) on the surface, which enabled the production of briquettes with a volume of 6.5 cm^3^. The gap between the rollers is 2 mm.

The essential part of this work includes the analysis of the stress distribution in the forming elements, the material flow in the pressing region, the briquettes density determination, and the energy-power parameters of the briquetting process. Considering the limited number of discrete elements that can be used for simulation, separate models were prepared to study specific parameters to achieve the best results.

One of the parameters that characterise the deformation region in a roller press is the amount of pressure exerted on the surfaces of the forming elements. Its value, considering the physical and mechanical properties of the compacted material, makes it possible to determine such parameters of the produced briquettes as density distribution and strength.

Specific difficulties arose when preparing a model for measuring the distribution of stresses in the forming cavities. Due to the large size of the model particles of the compacted material, the stresses arising on the working surfaces of the rollers are of a point nature, and their distribution is uneven. It is difficult to determine the magnitude and distribution of stresses on the pressing surfaces accurately. Therefore, it was decided to analyse various parts of the roller surfaces involved in the material compaction process separately (Figure 4). Isolating the forming elements from the rest of the work roller also focuses on the area most important for analysing the material compaction process. Each forming element was divided into 24 areas. It was assumed that the stresses in the formed briquette are symmetrical, making it possible to reduce the width of the formed element’s analysed surface to ⅗. The same approach was applied to the second work roller while highlighting the areas directly involved in forming briquettes.

The separation of areas on the surface of the groove also allows one to eliminate the stress that occurs at the place of separation of the briquettes. Because of the dimensions of the elements, which are larger than the gap between the rollers, the stress values obtained in these areas cannot be considered reliable.

Determination of stresses arising on each section’s surface consists of collecting information about the total compressive force acting on the surface of this section and then dividing it by the area of the section. The difficult moment of the analysis was the two corner elements of each cavity, which, due to the relatively small area, generated exceedingly high-stress amplitudes. The half briquette formation simulation carries out rollers’ power-required measurement for the briquettes formation. It is a necessary procedure, considering the limited number of discrete elements that can be used to simulate the compaction process. The pressing moment is determined when forming half of the briquette. Its value is multiplied by 2, then by the number of rows of forming elements (in this study 2), and again by 2, since 2 rollers are involved in the compaction. Therefore, the total multiplication value, in this case, is 8. Based on the determined value of the pressing moment and the rollers’ speed, the power of the system consumed during the formation of the briquettes is calculated.

The last measured parameter is the density of the obtained briquettes. Therefore, the model uses a geometry sensor placed in the briquette formation area so that each briquette leaving the forming element passes through this sensor. Using a cylindrical sensor allows one to remove particles of loose material that are not part of the briquette from the calculation (Figure 5).

Information about the mass of the compacted material is collected, which is currently in the centre of the specified sensor geometry. Briquette density is determined from the mass of the half of the briquette and the known total volume of the briquette 6.4 cm^3^ (the half of the briquette volume is 3.2 cm^3^).

## 3. Research Results and Analysis

### 3.1. The Simulation Results

Stress distribution and magnitude on forming element surfaces were determined for the studied materials (calcium hydroxide and a mixture based on copper ore concentrate). Calculations were made for the range of circumferential speed of rollers 0.1 ÷ 0.3 m/s. An example of the result of such calculations is shown in Figure 6.

Table 2 and Table 3 present stress characteristics on the forming elements’ working surfaces during the briquetting of calcium hydroxide and a mixture based on copper ore concentrate. Figure 7 and Figure 8 show the values of the roller’s torque and power consumption for the studied materials and the accepted characteristics of the pressing rollers working surfaces. Table 4 shows the calculated dependence of the density of briquettes on the value of the peripheral speed of the rollers.

The obtained results make it possible to analyse the studied materials’ compaction characteristics and configure the forming elements. Simulation of material compaction in a roller press made it possible to fix the distribution of stresses on the forming surfaces. The results clearly show a differentiated stress distribution between two surfaces: a recess and a groove (Figure 9).

On the cavity surface and the shaping groove, higher stresses occur in the upper part of the briquette, which coincides with the known density distribution of the briquettes. The exception is the lower section. Here, the stresses are much higher than in the previous layer. It is probably due to the rapid movement of the material at the last stage of briquette formation (Figure 10), and no material is allowed to escape from the mould cavity when the mould cavity is closed.

Differences in the stress distribution on the forming cavity surface and the groove are reduced to the maximum stress distribution on the briquette surface. For the forming cavity, the maximum values fall on the outer sections and take similar values along the entire height of the briquette. At this point, the very high-stress values coincide with the edge wear of the work roller surfaces seen in roller presses. On the other hand, on the groove surface, the maximum stresses for a given width fall on the axis of the briquette, and their maximum values occur only on its upper part. The stress distribution in the forming groove is much more uniform and has a smaller amplitude. For calcium hydroxide, the maximum stress amplitude on the groove surface is 30 MPa, and on the surface of the formed cavity, 45 MPa. For a mixture based on copper ore concentrate, the maximum amplitudes are 46 MPa and 59 MPa, respectively.

Simulation results for the compaction process of calcium hydroxide and a mixture based on copper ore concentrate show that the stresses distribution on the forming surfaces is constant for specific geometric parameters of the forming elements and does not depend on the properties of the material and the operating modes of the roller press (Figure 11).

Modelling performed for four values of the circumferential speeds of the rollers shows that the material rate of deformation does not directly affect the stress distribution for both calcium hydroxide and a mixture based on copper ore concentrate (Figure 11). This is very well illustrated by the average stress values on the surfaces of the forming elements. For calcium hydroxide, the average stress value range is 25÷28 MPa. A slight decrease in stresses was also observed with an increase in the circumferential speed of the rollers. Although this phenomenon occurs when materials are compacted in a roller press, the slight decrease in stress, in this case, can only be due to measurement errors. A similar dependence is not observed for a mixture based on copper ore concentrate. The average stress values are 36÷40 MPa, where the minimum value is determined for the circumferential speed of the rollers of 0.2 m/s (Figure 12). These results align with the experiments presented in [53,54], which demonstrated higher pressures and greater density of saddle-shaped briquettes in their upper part.

### 3.2. Comparison of Simulation Test Results with Experimental Data

An essential aspect of the study is comparing the results obtained by modelling using the discrete element method with experimental tests to verify the results obtained. First, the results of the average briquette density in the total volume obtained in the simulation and in the laboratory tests were compared (Table 5). The comparison of the results shows that in the case of laboratory tests, the briquette densities are slightly higher. The result can be considered satisfactory because the differences in the density of briquettes are, respectively, 8.8% for briquettes made of calcium hydroxide and 4.7% for briquettes made of copper ore concentrate. On this basis, it can be assumed that the simulation tests were carried out correctly.

Studies of the compactibility of a mixture based on copper ore concentrate on a roller press using rollers to produce saddle briquettes show that the maximum stress values for the moisture of 4.2% should be within 55 MPa (Figure 13). The higher values obtained in the experiment, up to 70 MPa, do not indicate errors in the developed model of material consolidation in the roller press. In experimental tests, the maximum values are measured only along the briquette axis. For comparison, therefore, the maximum stresses in the briquette axis in the simulation for copper ore concentrate are in the range of 41–66 MPa.

For calcium hydroxide, similar simulated stresses in the axis are 35÷41 MPa, while experimental tests of briquetting this material show maximum stresses in the range of 25÷50 MPa. Thus, there is a sufficiently good agreement between the simulation results of the actual compaction characteristics of the materials under study.

Other parameters recorded in the simulation of material briquetting on a roller press were the torque of the roller and power consumption (Figure 14). During the experiments, the roller torque during briquetting of calcium hydroxide and a mixture based on copper ore concentrate had linear characteristics. The difference between the torque values determined theoretically and those established experimentally does not exceed 60 Nm.

The results of experimental tests were used to analyse the power consumption. They included the briquetting of calcium hydroxide at peripheral speeds of the work rollers of 0.1, 0.2, and 0.3 m/s. A comparison of the power consumption values with the results of experimental studies shows some discrepancies with an increase in the strain rate. At the same time, a perfect convergence of the power consumption values is observed at a speed of 0.1 m/s, which is approximately 2 kW (Figure 14).

The experimental tests indicate a decrease in the strength of briquettes with an increase in the speed of the peripheral rollers. It was not observed in the simulation. This is due to the assumptions made in the model.

Experimental tests on the briquetting of a mixture based on copper ore concentrate covered only a speed of 0.15 m/s. For a saddle briquette, the recorded power consumption was around 6 kW (Figure 15). The power consumption for a speed of 0.15 m/s, determined in the simulation, was 5.34 kW.

A very important aspect of the work is to show that the briquette compaction is not uniform in the entire volume of the briquette. In places where the value of the unit peak is higher, there is a greater density of the material. The results of the obtained simulation correspond very nicely with the actual tests of the density of briquettes on their cross-section as well as the thermal imaging tests of the briquetting process of saddle briquettes carried out by the authors. In all cases, a higher briquette density is obtained in its upper part (Figure 16).

## 4. Conclusions

In this work, the DEM method was used for the first time to study the process of briquetting of fine fraction materials in roller presses with a complex configuration of forming elements. DEM is used to model the behaviour and determine the physical and mechanical properties (compaction coefficient, Young’s modulus, cohesion energy) of the studied materials—calcium hydroxide and cooper concentrate mixture.

For the studied materials, the character of distribution and size of contact stresses on surfaces of forming elements was established. The dependences of the change in the value of the average stress values on the change in the circumferential speed of the rolls were determined.

The connection between the circumferential speed of rotation of the rolls and the force parameters of the briquetting process was investigated. Graphical and analytical dependencies were established, which show the increase in rolling torque and energy consumption with increasing peripheral roll speed. It is noted that in the studied range (0.1÷0.3 m/s), the density of the obtained briquettes does not change significantly (not more than 3%). The average density of briquettes obtained in laboratory tests is higher than in simulation, respectively, 8.8% for briquettes made of calcium hydroxide and 4.7% for briquettes made of copper ore concentrate, which can be considered a satisfactory result.

An analysis of two material (calcium hydroxide and cooper concentrate mixture) models and four peripheral roller speeds (from 0.1 up to 0.3 m/s) showed that the stress distribution characteristic is mainly related to the geometry of the forming elements. For calcium hydroxide, the maximum stress amplitude on the groove surface is 30 MPa, and on the surface of the formed cavity, 45 MPa. For a mixture based on copper ore concentrate, the maximum amplitudes are 46 MPa and 59 MPa, respectively. Modelling performed for four values of the circumferential speeds of the rollers shows that the average stress values for calcium hydroxide are in the range of 25÷28 MPa, and for a mixture based on copper ore concentrate, 36÷40 MPa.

A complete model of material compaction in a roller press was developed, which makes it possible to determine the process’s power parameters and the material compaction’s characteristics, taking into account the characteristics of its constituent particles.

Discrete element modelling for the design of roller presses and material compaction processes can help a new configuration of forming elements development that provides rational characteristics of the briquetting process, taking into account the properties of the material to be briquetted and the operating modes of the press. Despite the accepted simplifications, in most cases, obtained results are in good agreement with the experimental data. A significant error was only found in the results of the power parameters determined for high values of the circumferential speed of the rollers when briquetting calcium hydroxide.

For further development of the material compaction model in a roller press, it is necessary to consider the influence of material moisture. Studies show that it significantly impacts the strength parameters of briquettes. An interesting problem in developing the model is the creation of methods that consider the actual particle size distribution of the material being briquetted and the conditions for the contact interaction of particles.

The presented results are a prerequisite for the discrete element method development for the phenomena occurring in roller presses description and to accelerate the design of press equipment and briquetting technological processes.

## Figures and Tables

**Figure 1 materials-15-04901-f001:**
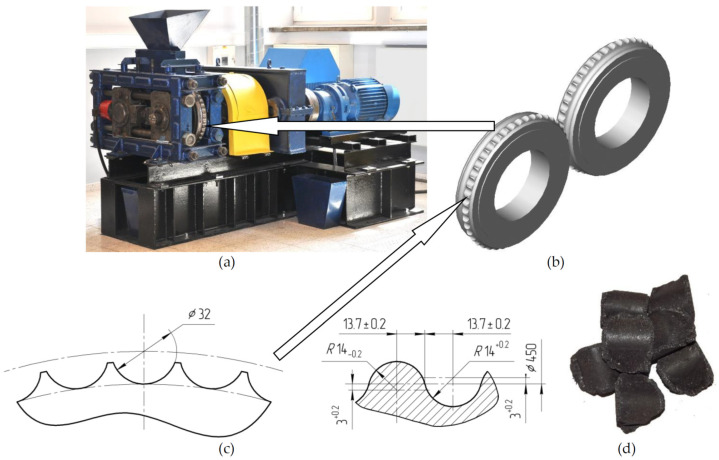
The equipment based on which the simulation tests were carried out: (**a**) the LPW450 roller press; (**b**) the rollers set for the saddle type briquette production; (**c**) the cross-section of the rollers; (**d**) the real saddle-shaped briquettes.

**Figure 2 materials-15-04901-f002:**
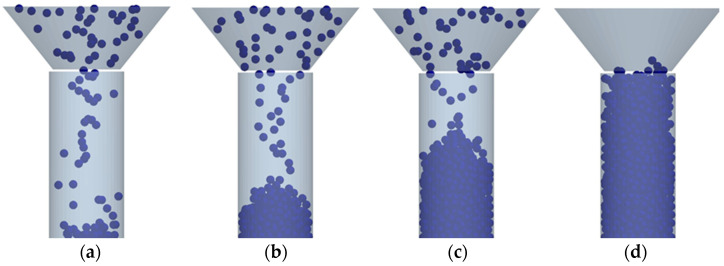
The course of cylinder filling modelling with bulk material to determine the density of its particles: (**a**–**d**) the next steps in the process.

**Figure 3 materials-15-04901-f003:**
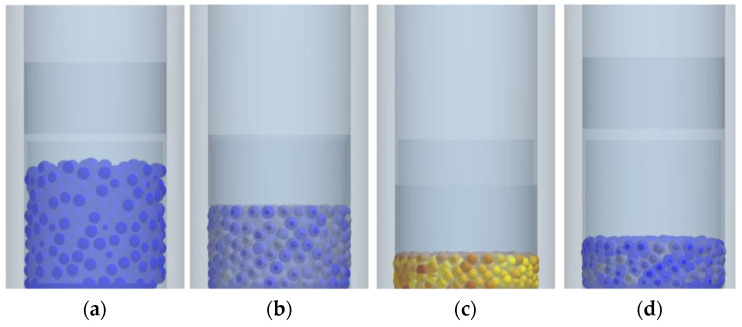
Simulation of the Young’s modulus determination for the compacted calcium hydroxide mixture: (**a**) material in the matrix without load; (**b**) material in the matrix partially loaded; (**c**) material in the matrix fully loaded; (**d**) material in the matrix unloaded.

**Figure 4 materials-15-04901-f004:**
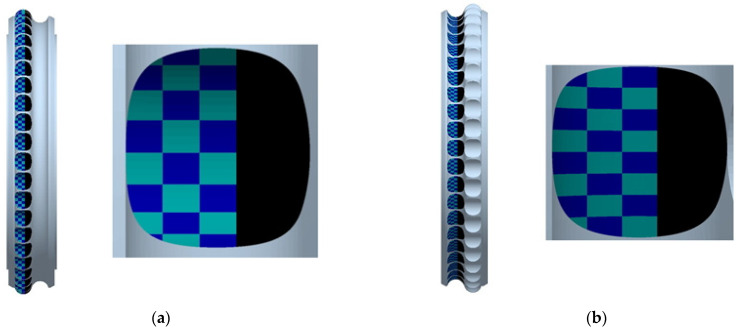
The location of the measuring sites on the surface of the forming elements: (**a**) cavities; (**b**) groove.

**Figure 5 materials-15-04901-f005:**
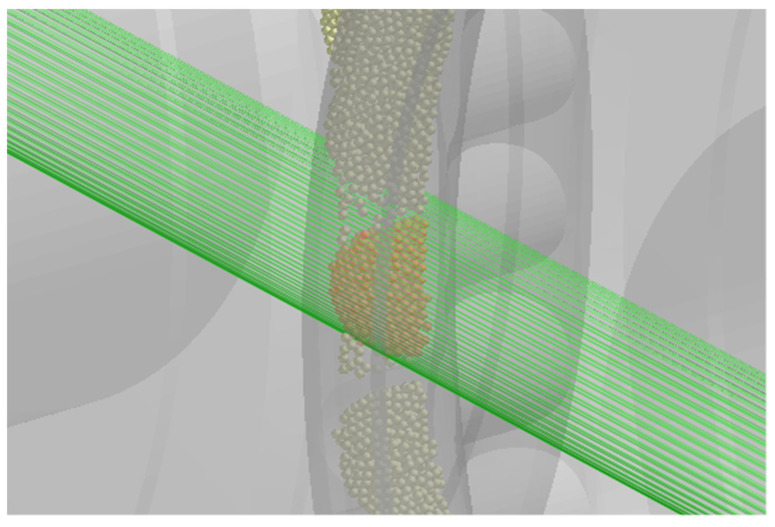
Location of the briquette weight measurement sensor in the model.

**Figure 6 materials-15-04901-f006:**
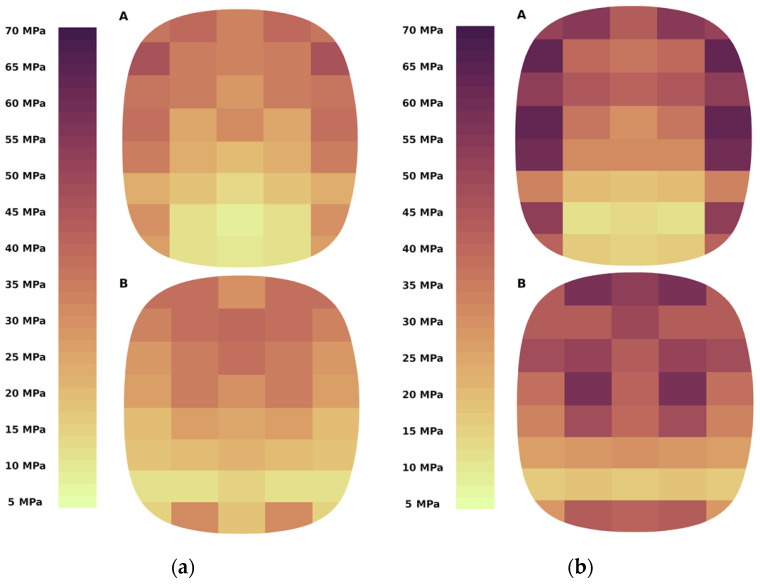
Stress distribution on forming surfaces for briquetting at the peripheral speed v = 0.1 m/s, A—cavity, B—groove: (**a**) calcium hydroxide; (**b**) copper ore concentrate.

**Figure 7 materials-15-04901-f007:**
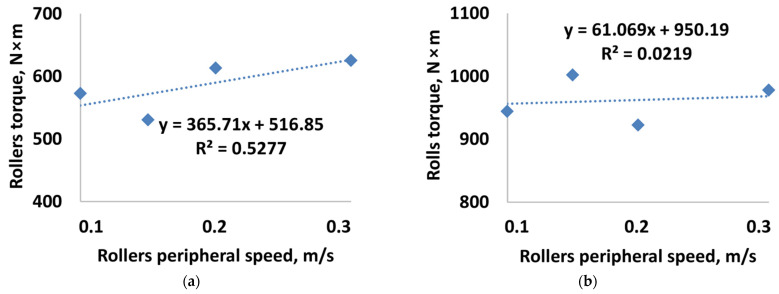
The dependence of rollers torque on the peripheral speed of the rollers when briquetting: (**a**) calcium hydroxide; (**b**) mixture based on copper ore concentrate.

**Figure 8 materials-15-04901-f008:**
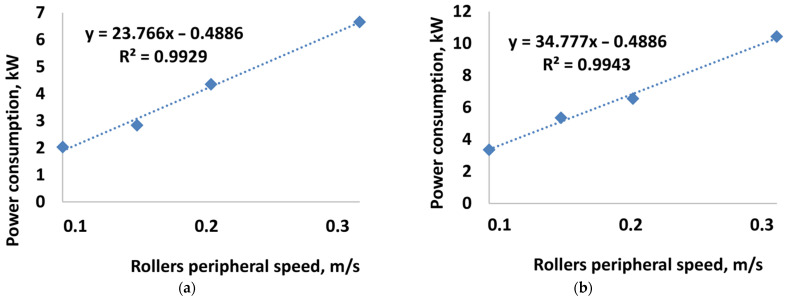
The dependence of power consumption on the peripheral speed of the rollers when briquetting: (**a**) calcium hydroxide; (**b**) mixture based on copper ore concentrate.

**Figure 9 materials-15-04901-f009:**
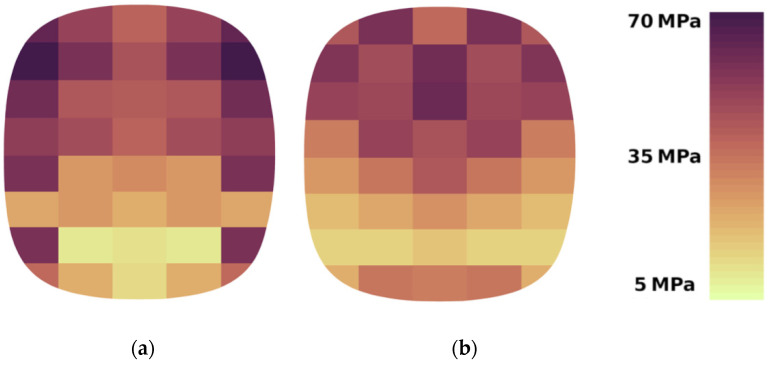
Differences in the stress distribution between the forming surfaces on copper ore concentrate briquetting at the peripheral speed of working rolls 0.3 m/s: (**a**) forming cavity; (**b**) forming groove.

**Figure 10 materials-15-04901-f010:**
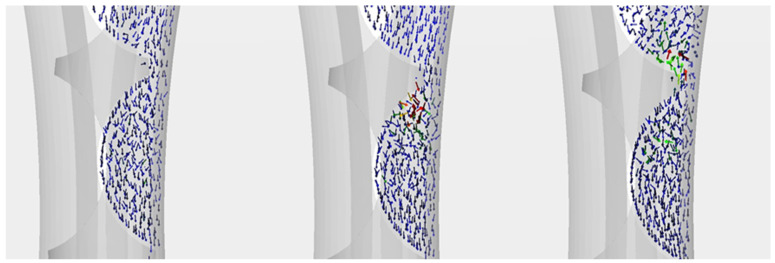
Distribution of material movement at the final stage of briquette production.

**Figure 11 materials-15-04901-f011:**
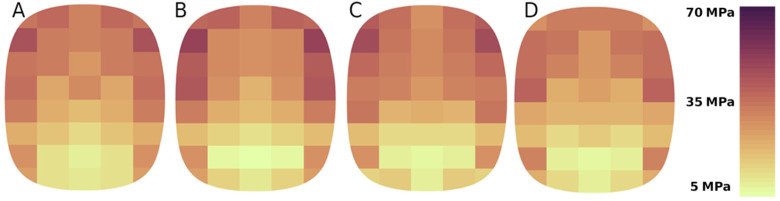
Stresses distribution on the forming cavity surface during briquetting of calcium hydroxide for the circumferential speed of the rollers: (**A**) 0.1 m/s, (**B**) 0.15 m/s, (**C**) 0.2 m/s, (**D**) 0.3 m/s.

**Figure 12 materials-15-04901-f012:**
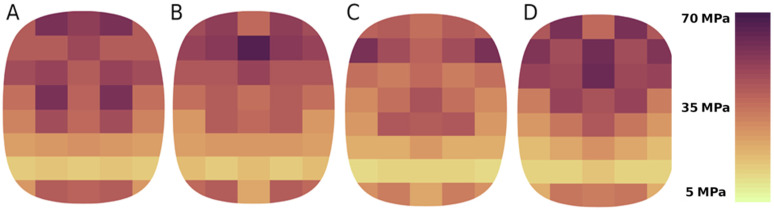
Stress distribution on the forming groove surface during briquetting of a mixture based on copper ore concentrate for circumferential speeds of rollers: (**A**) 0.1 m/s; (**B**) 0.15 m/s; (**C**) 0.2 m/s; (**D**) 0.3 m/s.

**Figure 13 materials-15-04901-f013:**
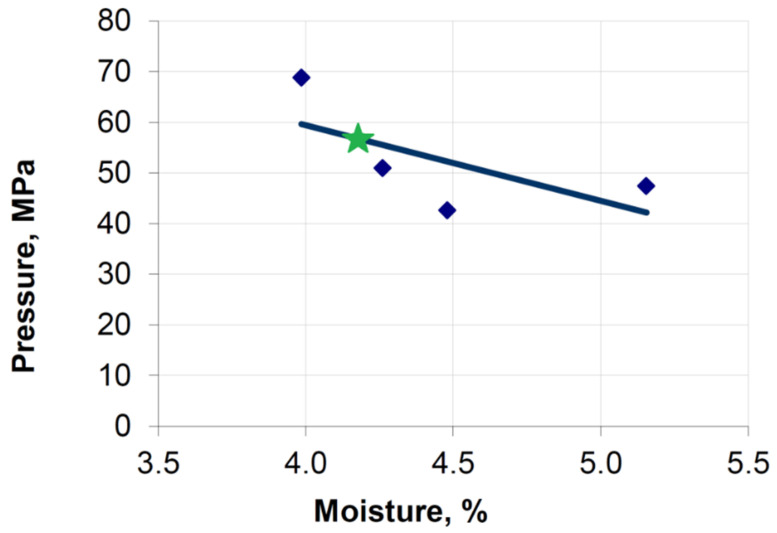
Influence of moisture content of raw materials on the specific pressure exerted on the bottom of the forming element when briquetting a mixture based on copper ore concentrate in a roller press [55]: the green star is for moisture 4.2%.

**Figure 14 materials-15-04901-f014:**
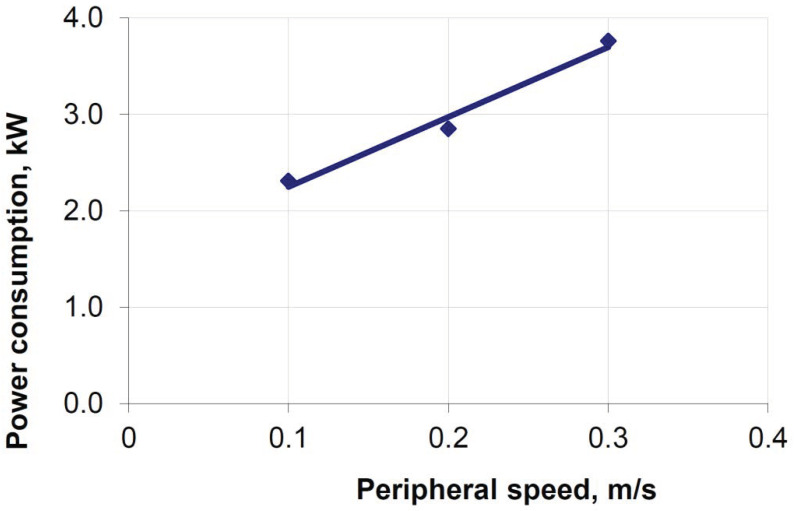
The dependence of the press drive power on the moisture content of the material and the circumferential speed of the rollers when briquetting calcium hydroxide [55].

**Figure 15 materials-15-04901-f015:**
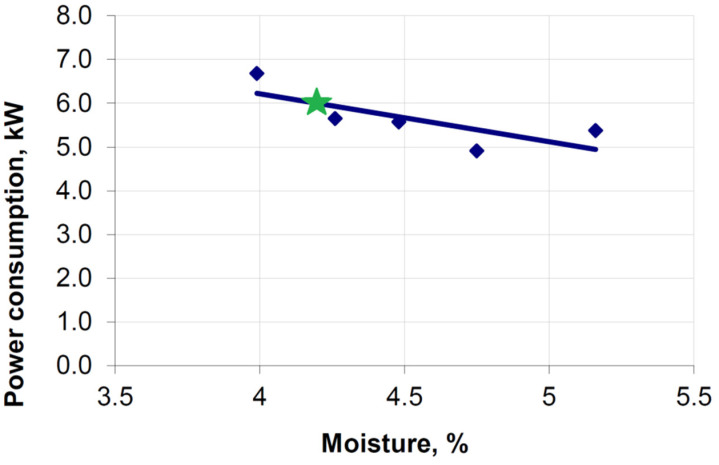
Influence of moisture content on energy consumption during briquetting of a mixture based on copper ore concentrate at peripheral speed of rollers *v* = 0.15 m/s [55]: the green star is moisture is 4.2%.

**Figure 16 materials-15-04901-f016:**
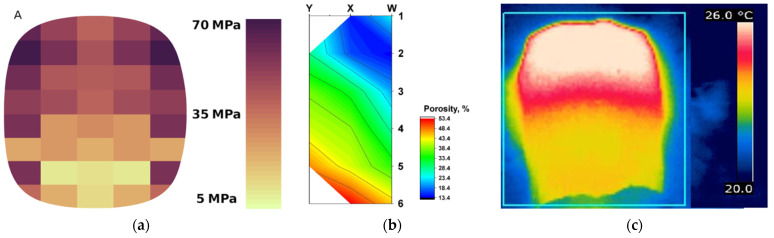
A general correlation between pressure distribution, porosity, and temperature distribution: (**a**) pressure distribution—DEM; (**b**) map of the porosity distribution on the copper briquette cross-section [53]; (**c**) the thermographic image of the briquette after briquetting [54].

**Table 1 materials-15-04901-t001:** Compaction characteristics of calcium hydroxide and mixture based on copper ore concentrate.

	Compaction Coefficient	Stress, MPa	Young’s Modulus, GPa	Cohesion Energy Density, J/m^3^
Calcium hydroxide	3.35	120.0	1.70	6.2 × 10^7^
Copper ore concentrate mixture	1.85	127.0	4.65	2.1 × 10^8^

**Table 2 materials-15-04901-t002:** Characteristics of stresses on the surfaces of the forming elements during briquetting of calcium hydroxide.

Rollers Peripheral Speed, m/s		Stress Max, MPa	Stress Min, MPa	Average Stresses, MPa	Average Stresses, MPa
0.10	cavity	51.03	5.47	27.84	28.42
	groove	39.37	12.58	29.00	
0.15	cavity	41.18	7.98	28.06	27.97
	groove	41.09	11.05	27.88	
0.20	cavity	46.46	7.77	27.71	27.39
	groove	37.90	11.54	27.06	
0.30	cavity	40.82	7.42	26.13	25.95
	groove	36.95	11.65	25.78	

**Table 3 materials-15-04901-t003:** Characteristics of stresses on the surfaces of the forming elements during briquetting of copper ore concentrate.

Rollers Peripheral Speed, m/s		Stress Max, MPa	Stress Min, MPa	Average Stresses, MPa	Average Stresses, MPa
0.10	cavity	64.63	12.33	38.65	39.34
	groove	59.66	17.01	40.04	
0.15	cavity	70.64	11.19	42.16	40.95
	groove	66.97	16.74	39.74	
0.20	cavity	63.53	10.86	37.78	36.52
	groove	58.55	14.39	35.26	
0.30	cavity	70.83	11.34	41.69	40.96
	groove	62.16	15.21	39.70	

**Table 4 materials-15-04901-t004:** The dependence of the average density of briquettes on the peripheral speed of the rollers.

Peripheral speed of the rollers, m/s	0.10	0.15	0.20	0.30
Average density of briquettes from calcium hydroxide, g/cm^3^	1.56	1.60	1.61	1.60
Average density of briquettes from mixture based on copper ore concentrate, g/cm^3^	2.457	2.416	2.425	2.434

**Table 5 materials-15-04901-t005:** Comparison of the average density of briquettes obtained in simulation and laboratory tests.

Peripheral speed of the rollers, m/s	0.10	0.15
Average density of briquettes from calcium hydroxide, g/cm^3^ simulation	1.56	1.60
Average density of briquettes from calcium hydroxide, g/cm^3^ experiment	1.71 ± 0.02	-
Average density of briquettes from mixture based on copper ore concentrate, g/cm^3^-simulation	2.46	2.42
Average density of briquettes from mixture based on copper ore concentrate, g/cm^3^-experiment	-	2.58 ± 0.08

## Data Availability

The data presented in this study are available upon request from the corresponding author.

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
