# Peer review of "Modelling of the Fine-Grained Materials Briquetting Process in a Roller Press with the Discrete Element Method"

_materials, 2022, doi:10.3390/ma15144901_

Round 1

Reviewer 1 Report

This current manuscript looks good. few issues should be paid attention to:

1) Full name for EDEM, there should be an explanation here.

2) Too many words on the three methods in the Introduction, including the Johanson method, the slab method and the finite element method, will make readers confused that why tons of words are used here, rather than your emphasis, EDSM. 

3) Some tenses or grammar were uncorrected. Please check the whole manuscript again for further submission.  

Author Response

Dear Reviewer 1

We are very grateful for your comments about the manuscript. Those comments are valuable and very helpful for improving our revised paper and the essential guiding significance to our research. We have carefully studied the comments and made corrections, which we hope to meet with approval.

Comment 1

Full name for EDEM, there should be an explanation here.

Answer: Thank you for the comment. It was corrected.

Comment 2

Too many words on the three methods in the Introduction, including the Johanson method, the slab method and the finite element method, will make readers confused that why tons of words are used here, rather than your emphasis, EDSM. 

Answer: Thank you for the comment. The Introduction was corrected and supplemented.

Comment 3

Some tenses or grammar were uncorrected. Please check the whole manuscript again for further submission.  

Answer: Thank you for the comment. The manuscript was corrected.

Reviewer 2 Report

Overall, this is a high-quality manuscript that is clear, concise, and well-written. The introduction is relevant and sufficient information about the previous study findings is presented for readers to follow the present study rationale and procedures. The method is appropriate, and the results are clear. The authors make a systematic contribution to the research literature in this area of investigation, useful contribution to the community, and adhere to the standard of the journal. I recommend accepting the paper for publication in the present form.

Author Response

Dear Reviewer 2,

Thank you very much for taking the time to read our manuscript thoroughly and make recommendations for its acceptance.

Reviewer 3 Report

The authors investigated the briquetting process using DEM simulations. The paper is on a topic of importance and will be of interest to others working in the field. I recommend publication with major changes.

1.    The authors should provide more detailed information on how to select the material properties in Table 1, which are critical for DEM simulations.

2.    For the density simulations, it seems that the solids fraction in the cylinder is only around 30%. For same size spheres, the packed solids fraction can be up to 65%. I suggest that the authors increase the cylinder size and rerun the simulations.

3.    The authors showed stress distribution in a few figures. I highly recommend that the authors add briquette density plots in these figures and see the correlations between the stresses and densities.

4.    The authors compared the simulations and experiments in Fig. 12 and 14 based on process conditions, such as pressure and power. It will be more interesting to see the comparison based on the product quality, such as the briquette density and strength. Products are the focus of customers.

Author Response

Dear Reviewer 3,

We are very grateful for your comments about the manuscript. Those comments are valuable and very helpful for improving our revised paper and the essential guiding significance to our research. We have carefully studied the comments and made corrections, which we hope to meet with approval.

Comment 1

The authors should provide more detailed information on how to select the material properties in Table 1, which are critical for DEM simulations.

Answer: The parameters were determined based on the parameters from the author's laboratory experimental studies of material compaction in a closed matrix described in the article:

Bembenek, M. Modeling The Variability Of The Virtual Young Module During The Consolidation Of Fine-Grained Materials. Molodyy Vcheny=Young Scientist. 2018, 12(64), 229-235, https://doi.org/10.32839/2304-5809/2018-12-64-57.

It was supplemented in the article.

Comment 2

For the density simulations, it seems that the solids fraction in the cylinder is only around 30%. For same size spheres, the packed solids fraction can be up to 65%. I suggest that the authors increase the cylinder size and rerun the simulations.

Answer: We agree with the reviewer; however, in our opinion, this relationship applies to fine-grained but not dusty materials. This relationship can be ideally used to calculate the bulk density e.g. material, after crushing in a jaw crusher (much larger grains). However, please remember that the bulk density of the material decreases as the grain is refined. According to the manufacturer of calcium hydroxide, its bulk density is 0.35-0.45kg/dm3

in our case an extra amount of water was added which slightly increases the bulk density.

According the http://www.microkat.gr/msdspd90-99/Calcium%20hydroxide.html

the density of solid Ca(OH)2 is 2.24 g/cm3.

The authors' own research on the bulk density shows that the mixture of copper ore concentrate with 5% sulphite lye with a moisture content of 4.2% prepared for briquetting has a bulk density in the range of 1.55 ± 0.04 g/cm3. Estimation of the real density of the solid copper ore concentrate is difficult, but assuming the share of copper in the concentrate 25% and the density of: copper 8.9 g/cm3 and the waste rock 2.7 g/cm3 can be estimated:

0.25*8.9 g/cm3 + 0.75*2.7 g/cm3 = 4.39 g/cm3

In our case, we believe that these parameters in the article have been entered correctly.

Comment 3

The authors showed stress distribution in a few figures. I highly recommend that the authors add briquette density plots in these figures and see the correlations between the stresses and densities.

Answer: Thank you very much for the remark. We want to present this topic in the following article.

Comment 4

The authors compared the simulations and experiments in Fig. 12 and 14 based on process conditions, such as pressure and power. It will be more interesting to see the comparison based on the product quality, such as the briquette density and strength. Products are the focus of customers.

Answer: Thank you very much for the remark. However, please remember that the article is aimed at specialists focused on that topic. With the pressure on the briquette shown, it is possible to estimate whether the briquette is consolidated correctly and additionally to estimate the hydraulic pressure system of the sliding (supporting) roller in the roller press. Having the estimated power, the parameters of the press drive system can be selected. It should be remembered that the strength parameters of briquettes change over time after their production. Typically, the strength of briquettes increases during the seasoning process. Depending on the requirements of briquette recipients, the technological process can also be set so that the briquettes meet the required strength parameters

Reviewer 4 Report

Remarks:

1.       To harmonize the text for the materials used in the methodology lines 124-126 and Table 1

2.       It is not clear why a vessel with a volume of 10 cm3 is used, since from the text lines 163-167 of the specified mass of material and its density is filled less than one-third of the cylinders ?!

3.       The reference to Fig. 1b and 1d (lines 193 and 194) may need to be understood 1a and 1b!

4.       The citations of Fig. 12 and 13 in my opinion are not sufficiently substantiated. It is sufficient to use only the obtained values from these sources in support of the authors' statements. Because the current simulation indicates the use of two humidity - lines 125 and 126 respectively. On the other hand, the data in fig. 12 illustrate the possibility of their approximation with a parabola equation from which it will be seen that the specific pressure has a minimum at a certain humidity because the minimum specific load leads to minimal energy costs. The idea of approximation with rights and determination on the basis of "maximum stress values should be within 40 ÷ 60 MPa" is not clear?!

5.       References 17, 33, 34, and 35 are not cited in the text. In the text, under Fig. 12 a source with the number 53 is cited but all references are 50!

6.       Tables 2 and 3 have the same titles but with different content!!! The data in the tables show that the deviation is a significant feature of the results obtained. This suggests that adding it to the tables will be useful.

7.       Not clear comment (line 349-351) -„During the experiments, the roller torque during briquetting of calcium hydroxide and a mixture based on copper ore concentrate had constant characteristics.!?

8.       The conclusions need to be reworked:

-          the second paragraph - to drop the first sentence;

- In the third paragraph - It is stated that the results obtained are in good agreement with the experimental data, after which (lines 392-393) it is stated „Despite the accepted simplifications, the obtained results are in good agreement with the experimental data.“, but in line 393-395 – “In the results of the power parameters determined for high values of the circumferential speed of the rollers when briquetting calcium hydroxide, a significant error was found

Author Response

Dear Reviewer 4,

We are very grateful for your comments about the manuscript. Those comments are valuable and very helpful for improving our revised paper and the essential guiding significance to our research. We have carefully studied the comments and made corrections, which we hope to meet with approval.

Comment 1

To harmonize the text for the materials used in the methodology lines 124-126 and Table 1

Answer: Thank you for the comment. It was corrected.

Comment 2

It is not clear why a vessel with a volume of 10 cm3 is used, since from the text lines 163-167 of the specified mass of material and its density is filled less than one-third of the cylinders ?!

Answer: Thank you for the comment. This is because we are handling here with two types of density - bulk and real density. The bulk density (of granular material with air spots between grains) is always lower, even several times than the real density (of a solid material).

Comment 3

The reference to Fig. 1b and 1d (lines 193 and 194) may need to be understood 1a and 1b!

Answer: Thank you for the comment. This has been corrected and detailed on lines 112-115.

Comment 4

The citations of Fig. 12 and 13 in my opinion are not sufficiently substantiated. It is sufficient to use only the obtained values from these sources in support of the authors' statements. Because the current simulation indicates the use of two humidity - lines 125 and 126 respectively.

Answer: Thank you for the comment. The figures in the article have been corrected.

On the other hand, the data in fig. 12 illustrate the possibility of their approximation with a parabola equation from which it will be seen that the specific pressure has a minimum at a certain humidity because the minimum specific load leads to minimal energy costs.

Answer: Our many years of experience show, however, that in the case of briquetting the copper ore concentrate mixture, the pressures decrease with the increase in the moisture of the mixture. It is caused by the material becoming a plastic body, until a high water content causes sticking of the mould cavities. Only the forming process then takes place, not the briquetting process. Then the energy consumption for the process is minimal.

The idea of approximation with rights and determination on the basis of "maximum stress values should be within 40 ÷ 60 MPa" is not clear?!

Answer: Thank you for the comment. This has been changed and detailed in the article

Comment 5

References 17, 33, 34, and 35 are not cited in the text. In the text, under Fig. 12 a source with the number 53 is cited but all references are 50!

Answer: Thank you for the comment. We apologize for these editorial shortcomings. This has been corrected in the text

Comment 6

Tables 2 and 3 have the same titles but with different content!!! The data in the tables show that the deviation is a significant feature of the results obtained. This suggests that adding it to the tables will be useful.

Answer: Thank you for the comment. We apologize for these editorial shortcomings. This has been corrected in the text

Comment 7

Not clear comment (line 349-351) -„During the experiments, the roller torque during briquetting of calcium hydroxide and a mixture based on copper ore concentrate had constant characteristics.“!?

Answer: Thank you for the comment. We apologize for these editorial shortcomings. This has been corrected in the text

Comment 8

The conclusions need to be reworked:

-  the second paragraph - to drop the first sentence;

- In the third paragraph - It is stated that the results obtained are in good agreement with the experimental data, after which (lines 392-393) it is stated „Despite the accepted simplifications, the obtained results are in good agreement with the experimental data.“, but in line 393-395 – “In the results of the power parameters determined for high values of the circumferential speed of the rollers when briquetting calcium hydroxide, a significant error was found“

Answer: Thank you for the comment. We apologize for these editorial shortcomings. This has been corrected in the text

Reviewer 5 Report

This study concentrates on the application of the discrete element method for modeling the process of roller compaction and press. For this purpose, a computer software (EDEM) is used to develop models of material compaction considering calcium hydroxide and copper ore concentrate. 

The paper is well-written and organized, and the topic is interesting, important, and falls within the scope of the journal. Therefore, I would like to suggest its publication provided that the following comments are considered by the authors to further improve its quality:

-       The introduction section is very general. It is suggested to mention the advantages of developing models using DEM compared with conventional FEM, to show the merits of the performed study.

-       The same goes for the conclusion section. Instead of mentioning general conclusions such as the DEM approach can be used for modeling this phenomenon or some similar statements, it is required to mention a quantitative summary of the results attained from the performed analysis using the developed model. 

-       Although, understandably, this is the authors’ field of research and they surely have previous publications, the references include many self-citations (about 20% of the references). It is suggested to improve the literature review by addition of more related references. 

-       The research significance is missing. In the last paragraph of the introduction, the authors should mention that what is the advantage of their study compared to previously published papers and what is the contribution of this research. 

-       Page 4, line 138: a suitable reference should be provided for the Hertz-Mindlin model. 

-       It is better to adopt a uniform unit system (CGS or MKS) in the text.

-        The accuracy of the developed model is verified using experimental results from published literature. However, the verifications are presented in the text in different locations. It would be better to add a subsection discussing verification of the attained results and the validity of the model. 

Author Response

Thank you very much for all the comments and suggestions. We did our best to implement all the suggested changes.

Remark 1

The introduction section is very general. It is suggested to mention the advantages of developing models using DEM compared with conventional FEM, to show the merits of the performed study.

Answer: Thank you for the comment. The Introduction was corrected and supplemented.

Remark 2

The same goes for the conclusion section. Instead of mentioning general conclusions such as the DEM approach can be used for modelling this phenomenon or some similar statements, it is required to mention a quantitative summary of the results attained from the performed analysis using the developed model.

Answer: Thank you for the comment. The Conclusion was corrected and supplemented.

Remark 3

Although, understandably, this is the authors’ field of research and they surely have previous publications, the references include many self-citations (about 20% of the references). It is suggested to improve the literature review by addition of more related references. 

Answer: Thank you for the comment. It has been corrected in the text. A new references has been added.

Remark 4

The research significance is missing. In the last paragraph of the introduction, the authors should mention that what is the advantage of their study compared to previously published papers and what is the contribution of this research.

Answer: Thank you for the comment. It has been suplemented in the text.

Remark 5

Page 4, line 138: a suitable reference should be provided for the Hertz-Mindlin model. 

Answer: Thank you for the comment. It has been added in the text.

Remark 6

It is better to adopt a uniform unit system (CGS or MKS) in the text.

Answer: In our article we used SI unit system which is recommended.

https://en.wikipedia.org/wiki/International_System_of_Units

Of course, we can change the system of units to another, but we consider it as standard in our research.

Remark 7

The accuracy of the developed model is verified using experimental results from published literature. However, the verifications are presented in the text in different locations. It would be better to add a subsection discussing verification of the attained results and the validity of the model.

Answer: Thank you for the comment. This has been corrected in the article and the new section has been developed.

Round 2

Reviewer 3 Report

Unfortunately, the authors didn't address the majority of my concerns. I highly recommend that the authors show at least one comparison of the simulation prediction and experimental data based on the briquette properties to validate the model. 

I agree that for dusty materials the solids fraction is much lower than 65%. However I record that the particle size used in your simulation is 2mm (please correct me if I was wrong). How did you justify that you simulate a dust system?  

Author Response

Thank you very much one again for all the comments and suggestions. We did our best to implement all the suggested changes.

Remark 1

Unfortunately, the authors didn't address the majority of my concerns. I highly recommend that the authors show at least one comparison of the simulation prediction and experimental data based on the briquette properties to validate the model.

Answer: Thank you for your attention. The density of the briquette has no constant value in the entire volume. In line with the remark, we compared the pressure distribution obtained in the DEM simulation with other studies conducted by us, the porosity distribution and the thermal imaging tests (Figure 16). In the case of thermography results, the higher the temperature on the briquette surface in a given place, the higher the pressure exerted in that place. All results are consistent regardless of the test method.

Remark 2

I agree that for dusty materials the solids fraction is much lower than 65%. However I record that the particle size used in your simulation is 2mm (please correct me if I was wrong). How did you justify that you simulate a dust system?

Answer: Dear Reviewer. We have been dealing with briquetting in roller presses for 20 years. We did a lot of experiments in the laboratory and in the technological lines. We designed roller briquetting machines which work in industry and we selected compaction units for different material parameters. We understand the topic very well. We are aware of the shortcomings of our article, please remember that we did not have the full version of the program. However, what we did, we showed that the briquetting process in the roller press can be modelled thanks to DEM, no one has done it before. So a new research field opens up. We are pioneers and we would like to develop this topic further. This topic took us over a year of research. Please treat our research as something new, to show a certain research possibility, developed despite very large limitations in a way that gives results consistent with laboratory research. We know that there is still a lot to be done in this regard. We would be grateful if you accept our manuscript for publication. It will surely help us to raise funds for future research for this purpose.

Round 3

Reviewer 3 Report

Dear authors, I used EDEM to simulate powder mixing and tableting more than 10 years ago, and have experience on roller compactor more than 20 years.  Validation is the key for any DEM simulations. 

I understand that the density of ribbons/briquettes are not uniform and the comparison between the model and experiments could be poor now. This is fine and could be improved in the future work. I don't understand that why you are so hesitant to show the comparison since you already have the data from both experiments and simulations. 

Author Response

Dear Reviewer,

Thank you very much once again for your review of our article and your patience!

Remark 1

Dear authors, I used EDEM to simulate powder mixing and tableting more than 10 years ago, and have experience on roller compactor more than 20 years. Validation is the key for any DEM simulations. 

I understand that the density of ribbons/briquettes are not uniform and the comparison between the model and experiments could be poor now. This is fine and could be improved in the future work. I don't understand that why you are so hesitant to show the comparison since you already have the data from both experiments and simulations. 

Answer

The validation of the density results was added to the article. The average density of briquettes obtained in laboratory tests are higher than in simulation respectively 8.8% for briquettes made of calcium hydroxide and 4.7% for briquettes made of copper ore concentrate which authors considered a satisfactory result.